# Plasma Membrane Potential of *Candida albicans* Measured by Di-4-ANEPPS Fluorescence Depends on Growth Phase and Regulatory Factors

**DOI:** 10.3390/microorganisms7040110

**Published:** 2019-04-24

**Authors:** Jakub Suchodolski, Anna Krasowska

**Affiliations:** Department of Biotransformation, Faculty of Biotechnology, University of Wroclaw, 50-383 Wrocław, Poland; jakub.suchodolski@uwr.edu.pl

**Keywords:** *Candida albicans*, plasma membrane potential, membrane polarization, di-4-ANEPPS, detergents

## Abstract

The potential of the plasma membrane (Δψ) regulates the electrochemical potential between the outer and inner sides of cell membranes. The opportunistic fungal pathogen, *Candida albicans,* regulates the membrane potential in response to environmental conditions, as well as the physiological state of the cell. Here we demonstrate a new method for detection of cell membrane depolarization/permeabilization in *C. albicans* using the potentiometric zwitterionic dye di-4-ANEPPS. Di-4-ANEPPS measures the changes in the cell Δψ depending on the phases of growth and external factors regulating Δψ, such as potassium or calcium chlorides, amiodarone or DM-11 (inhibitor of H^+^-ATPase). We also demonstrated that di-4-ANEPPS is a good tool for fast measurement of the influence of amphipathic compounds on Δψ.

## 1. Introduction

Though the plasma membrane potential (Δψ) is an electrochemical potential difference between extracellular and intracellular compartments in all living cells, the mechanisms maintaining Δψ differ between cell types [1]. Δψ acts as an indicator of the physiological status of the cell; for example, depolarization of the cell membrane in lymphocytes prevents cell proliferation [2]. The influence of the value of Δψ on the lipid lateral localization in the plasma membrane of the yeast *Saccharomyces cerevisiae* is another example that highlights the importance of the Δψ in cell biology [3].

*Candida albicans* is a microorganism of human microflora (skin, as well as urinary and gastrointestinal tracts) and the most common cause of opportunistic fungal infections of immunocompromised patients [4]. The value of *C. albicans* Δψ is ~−120 mV and is comparable to that of pathogenic bacteria, which ranges from ~−130 mV to ~−150 mV [5,6]. Unlike in *C. albicans*, the value of Δψ of non-pathogenic *S. cerevisiae* is ~−71 mV and is comparable to the potential of mammalian cells, which is ~−90 mV [5,6]. 

Highly desirable activities of antifungal compounds include binding to ergosterol and subsequent permeabilization of the cell membrane [7,8]. The loss of cell membrane integrity due to the action of antifungal drugs causes plasma membrane depolarization [9]. 

Two types of fluorescent probes are commonly used for measurements of *C. albicans* plasma membrane polarity: slow response potential-sensitive cationic carbocyanines (Dil, DiS and DiO) and anionic bis-barbituric acid oxonols (DiBAC) [10,11,12,13].

Carbocyanines accumulate in hyperpolarized membranes, while bis-oxonol dyes enter depolarized cells [13,14]. Binding to the cell by both groups of dyes results in a red shift of the fluorescence spectrum while a blue shift of fluorescence spectrum is observed when probes are not bound [12,15]. Accumulation of the cationic and anionic dyes in the plasma membrane and changes of Δψ caused by interfering factors require constant monitoring of the time course of the fluorescence spectrum shifts. Additionally, carbocyanines are substrates for *C. albicans* drug ATP-binding cassette (ABC) transporters (Cdr1 and Cdr2) and are used to measure the activity of these pumps in real time [15]. However, Cadek et al. [16] found that the excretion of carbocyanines by ABC transporters could interfere with the proper measurement of cell membrane potential.

Potentiometric zwitterionic aminonaphthylethenylpyridinium (ANEP) dyes (di-4-ANEPPS and di-8-ANEPPS) were previously used to map the membrane potential along neurons and muscle fibers [16,17,18]. Both probes reduce the excitation fluorescence intensity at ~440 nm and increase it at ~530 nm in response to membrane hyperpolarization [19,20]. In addition, after excitation in the range of ~470 nm to 490 nm, ANEP dyes cause a blue or red fluorescence shift during depolarization or hyperpolarization of membranes, respectively [21,22,23]. Di-4-ANEPPS was also used for measuring membrane potential in *S. cerevisiae.* The use of this dye in these walled cells showed its lower stability, but faster response, in comparison to previously used cationic and anionic dyes [24].

In this study, we report a new application of monitoring di-4-ANEPPS fluorescence spectral shift in *Candida albicans’* Δψ measurement. We developed a straightforward and reliable assay in monitoring de-/hyperpolarization as a result of ion homeostasis disturbance and after treatment with amphipathic compounds, which may provide a better understanding of the physiology of *C. albicans*.

## 2. Materials and Methods

### 2.1. Chemicals, Strains and Growth Conditions

All chemicals and reagents used in this study were purchased from the following sources: 3,3′-Dipropylthiacarbocyanine iodide (diS-C_3_(3)), sodium dodecyl sulfate (SDS), benzalkonium chloride (BAC), Triton X-100 (Sigma-Aldrich; Poznań, Poland); D-glucose, bacteriological agar, propidium iodide (PI) (manufacturer: Bioshop, distributor: Lab Empire; Rzeszów, Poland); peptone, yeast extract (YE) (manufacturer: BD; distributor: Diag-med; Warszawa, Poland); potassium chloride (KCl) (Chempur; Piekary Śląskie, Poland); calcium chloride (CaCl_2_) (POCH; Gliwice, Poland); pyridinium, 4-(2-(6-(dibutylamino)-2-naphthalenyl)ethenyl)-1-(3-sulfopropyl), hydroxide inner salt (di-4-ANEPPS) (Thermo Fisher; Warszawa, Poland); 2-dodecanoyloxyethyl-dimethylammonum chloride (DM-11) was a gift from Dr. Łuczyński (Wroclaw, Poland).

*C. albicans* strain CAF2-1 (genotype: *ura3∆::imm434/URA3*) was a kind gift from prof. D. Sanglard (Lausanne, Switzerland) [25]. It was routinely grown at 28 °C on YPD medium (2% glucose, 1% peptone, 1% YE) in a shaking incubator (120 rpm). Agar in a final concentration of 2% was used for medium solidification. 

To determine growth phases, CAF2-1 was grown in 20 mL of YPD medium for 24 h at 28 °C with shaking (120 rpm). Every two hours, the A_600_ measurements were performed using a Hach Odyssey DR/2500 spectrophotometer in three independent repetitions.

For specific experiments, CAF2-1 cells were grown until they reached either early (8 h) or late (14 h) logarithmic phase.

### 2.2. DiS-C_3_(3) Assay

The fluorescence assay of Δψ was performed in the early and late logarithmic phase of *C. albicans* growth as described previously [26]. Δψ measurements using de- and hyperpolarizing compounds (200 mM KCl, 50 μM DM-11; 25 mM CaCl_2_, 2 μM Amiodarone, respectively) and di-4-ANEPPS (final conc. = 3 x 10^−6^ M) were performed in the early phase of growth. All reagents were prepared shortly before fluorescence measurements and added at t = 0 min (de- and hyperpolarizing compounds) or at t = 60 min (di-4-ANEPPS).

### 2.3. Di-4-ANEPPS Assay

Detection of Δψ by di-4-ANEPPS was performed by labelling 3 mL of *C. albicans* cell suspensions (A_600_ = 0.1) in citrate phosphate (CP) buffer (pH 6.0). The final concentration of di-4-ANEPPS probe was 3 x 10^−6^ M. Samples were incubated for 30 min at room temperature (RT). The growth-dependent membrane potential was measured both in the early and late logarithmic phase of *C. albicans* growth. Membrane potential measurements using de- and hyperpolarizing compounds (200 mM KCl, 50 μM DM-11; 25 mM CaCl_2_, 2 μM Amiodarone respectively) were performed only in the early phase of growth because of physiological depolarization of plasma membrane in late log phase cells. KCl, DM-11, CaCl_2_, Amiodarone were added immediately after incubation of cells with di-4-ANEPPS. In all experiments, di-4-ANEPPS was excited at 488 nm (Ex slit = 10 nm) and fluorescence spectra at 520–720 nm (Em slit = 2.5 nm) (PMT voltage = 700 V) were recorded using fluorescence spectrophotometer (HITACHI F-4500) equipped with a xenon lamp. Each experiment was performed in three independent replications and each probe was excited three times. Fluorescence spectra from corresponding experiments were averaged and normalized (value 1 for maximum emission intensity in each case) for comparison of fluorescence maxima shifts.

### 2.4. Toxicity of di-4-ANEPPS

*C. albicans* suspensions were treated as described in Section 2.3 (CP buffer; A_600_ = 0.1; 3 x 10^−6^ M di-4-ANEPPS; 30 min), washed with CP, and resuspended in CP (10^0^). Suspensions were serially diluted up to 10^−3^, then 2 µL were spotted onto YPD agar and cultured for 48 h at 28 °C. Afterwards, the plates were photographed using FastGene^®^ B/G GelPic imaging box (Nippon Genetics, Dueren, Germany).

### 2.5. Influence of Detergents on Δψ

The impact of SDS (0–320 µg/mL), BAC (0–320 µg/mL) and Triton X-100 (0–320 µg/mL) on Δψ was evaluated as described in Section 2.3, with modifications. Fluorescence spectra of di-4-ANEPPS (3 x 10^−6^ M) solution in CP buffer (pH 6.0) were collected after 30 min incubation in the presence of detergents. Because an interaction between the fluorescent probe and detergents was identified, early log phase *C. albicans* cells were pretreated with detergents for 30 min at RT, washed three times with CP buffer (pH 6.0), and labelled with di-4-ANEPPS for 30 min. Fluorescence measurements were performed as described above. For data analysis, the red-blue signal ratio (R-B ratio) was calculated by dividing the sum of fluorescence intensity (IF) between 580 and 620 nm by the sum of IF between 540 and 580 nm as shown in the formula below.
(1)$$RB\;ratio=\frac{\sum_{i=580\;nm}^{620\;nm}IFi}{\sum_{i=540\;nm}^{580\;nm}IFi}$$

Additionally, all results were normalized to the control (value = 1 for the control experiment without the addition of detergents). In this approach, it was assumed that the fluorescence spectra symmetry had a maximum at 580 nm (plasma membrane potential of early log phase cells in control conditions); therefore, blue shift (depolarization) and red shift (hyperpolarization) result in an R-B ratio of <1 and >1, respectively.

### 2.6. Propidium Iodide (PI) Staining

Assessment of plasma membrane permeability was performed as described before [8], with modifications. Briefly, 3 mL of *C. albicans* cell suspensions (A_600_ = 0.1) in CP buffer (pH 6.0) were mixed with SDS (0–320 µg/mL), BAC (0–320 µg/mL) or Triton X-100 (0–320 µg/mL), incubated for 30 min at RT, washed three times with CP buffer, and stained for 5 min with PI to the final dye concentration of 6 x 10^−6^ M. Next, cell suspensions were washed twice with CP buffer and observed under a Zeiss Axio Imager A2 microscope equipped with a Zeiss Axiocam 503 mono microscope camera and a Zeiss HBO100 mercury lamp. The percentage of permeabilized cells was evaluated by counting PI positive cells out of one hundred cells in three independent repetitions for each experiment. Statistical significance analysis was performed using Student’s t-test (binomial, unpaired).

### 2.7. Sequences Alignmets

TOK1 gene and Tok1p sequences from *S. cerevisiae* S288C strain were obtained from *Saccharomyces* Genome Database (accession ID: SGD:S000003629) [27]. TOK1 gene and Tok1p sequences from *C. albicans* SC5314 strain were obtained from *Candida* Genome Database (accession by systematic name: C4_00670W_A) [28]. Sequences alignments were performed by EMBOSS Needle program [29].

## 3. Results and Discussion

Di-4-ANEPPS dye is widely used in the measurement of the Δψ in tissues [30], but in walled cells it was used only in yeast *S. cerevisiae* [24]. H^+^-ATPase forms Δψ in both pathogenic *C. albicans* and non-pathogenic *S. cerevisiae*, but its activity in these two species is different. In contrast to *S. cerevisiae*, *C. albicans* up-regulates energy reserve metabolism [31] and has a lower acidification activity of H^+^-ATPase [32]. Our previous investigations indicated that the Δψ of *C. albicans* measured by diS-C_3_(3) dye differs from *S. cerevisiae* [15]. To expand this observation, we used di-4-ANEPPS to measure *C. albicans*’ Δψ under different conditions and compare this method with the method using diS-C_3_(3).

### 3.1. Di-4-ANEPPS and DiS-C_3_(3) Measure Cell Depolarization Depending on the Phases of Growth

First, we compared diS-C_3_(3) and di-4-ANEPPS in the detection of growth phase-dependent plasma membrane depolarization for *C. albicans* (Figure 1A,B). Previously, depolarization of *S. cerevisiae* plasma membrane resulting from decreased H^+^-ATPase activity in the late exponential phase was observed using diS-C_3_(3) [16,33]. In the case of *C. albicans*, log phase was observed between 8 to 14 h of growth (Figure 1C). Staining with diS-C_3_(3) in the cells was considerably slower in the late log phase (λmax = ~ 572 nm at 40 min) compared to the early phase (λmax= ~ 577–578 nm at 28 min) of exponential growth (Figure 1A), which indicates membrane depolarization and is in agreement with our previous studies [15]. 

In our study, we used the *C. albicans* CAF2-1 strain, which has both ABC transporters (Cdr1 and Cdr2). DiS-C_3_(3) is the substrate for ABC transporters and its efflux is observed at 40–50 min after addition of it to the *S. cerevisiae* cell suspension [16]. In the case of *C. albicans,* an efflux of this dye occurs after 60–70 minutes (Figure 1E, control) [15]. For Δψ measurements, we did not monitor diS-C_3_(3) fluorescence longer than that to avoid ABC transporters interference.

We compared the fluorescence spectrum of di-4-ANEPPS bound to plasma membrane of *C. albicans* cells in early and late exponential phases of growth (Figure 1B). The di-4-ANEPPS emission maximum (EM) was at ~580 nm for the early exponential phase, whereas in the late exponential cells the EM shifted to ~574 nm, with the spectrum area being noticeably narrowed. Blue-shift of di-4-ANEPPS fluorescence indicates depolarization of the plasma membrane, as previously reported for tissues stained with ANEP dyes [23,34]. 

In our experiments, di-4-ANEPPS treatment was not toxic in *C. albicans* (Figure 1D). We conclude that Δψ measurements were not affected by an adverse effect of the dye on *C. albicans* cells. Additionally, di-4-ANEPPS did not inhibit diS-C_3_(3) efflux after addition at 60 min. (Figure 1E). The addition of ABC transporter substrate during diS-C_3_(3) assay results in lower or lack of diS-C_3_(3) efflux [15], thus di-4-ANEPPS is not an ABC transporter substrate and Δψ measurements were not affected by the ABC transporters efflux activity.

### 3.2. Di-4-ANEPPS and DiS-C_3_(3) Measure Cell Depolarization and Hyperpolarization Induced by External Factors

In addition to H^+^-ATPase, other transmembrane transporters form Δψ in pathogenic and non-pathogenic yeast. K^+^ ions are transported to the inside of the *C. albicans* cells by Trk1p uniporter. The single Trk isoform (CaTrk1p) in *C. albicans* is nearly 60% homologous in four transmembrane motifs to both isoforms of Trkp in *S. cerevisiae*; this is expected to reflect quantitatively similar functions of these transporters [35]. *C. albicans* needs a highly efficient K^+^ uptake system because of low concentration of potassium in the niches of the host organism [36]. The accumulation of potassium ions inside the cell depolarizes the plasma membrane, as demonstrated in *S. cerevisiae* by Gaskova et al. [37] using diS-C_3_(3) dye. We used diS-C_3_(3) and di-4-ANEPPS dyes to measure the *C. albicans* Δψ after KCl application (Figure 2A,B). A blue shift of λmax (di-4-ANEPPS; EM: 575 nm) and fluorescence intensity kinetics (diS-C_3_(3); λmax = ~ 575-576 nm at 50 min) were observed (Figure 2A,B), which indicate depolarization of the membrane.

Among transmembrane transporters that contribute to Δψ are regulators of intracellular potassium concentrations, such as the Tok1 channel [38]. Tok1p is a potassium specific channel that releases K^+^ from the cell and thus regenerates Δψ [39]. Deletion of the TOK1 gene results in the depolarization of plasma membrane, and conversely, its overexpression leads to hyperpolarization of the yeast plasma membrane [40,41]. In our investigations, the *C. albicans* Δψ was measured with diS-C_3_(3) dye in real time for 60 min. The Δψ grew more slowly after using KCl (λmax at ~50 min) than in cells not treated with KCl (λmax at ~30 min), but it finally achieved similar λmax values (Figure 2A). By using the di-4-ANEPPS dye and observing the blue shift of fluorescence intensity (control EM: 582 nm; KCl treated EM: 576 nm) we have shown that the plasma membrane of *C. albicans’* cells is depolarized (Figure 2B). However, the intensity of this depolarization is lower than after treatment of cells with DM-11 (EM: 574 nm), which is a known H^+^-ATPase inhibitor in yeast (Figure 2B) [42,43]. We observed reduced staining of cells with diS-C_3_(3) after 10 min. incubation with DM-11 and no Δψ recovery (Figure 2). DM-11 is a lysosomotropic agent whose deprotonated form penetrates membranes and protonated form accumulates in acidic yeast compartments (e.g., vacuoles). If this compound is used in high concentrations it can cause membrane disruption [44]. Zahumensky et al. [45] have observed the increase of Tok1p activity in *S. cerevisiae* cells treated with DM-11 and a gradually more extensive Tok1 channel activity with deeper depolarization of the membrane.

Our results indicate a weaker role of Tok1p in Δψ recovery after treatment of cells with DM-11 and deeper membrane depolarization than when using KCl (Figure 2A,B). According to the sequence alignment (Section 2.7), the CaTOK1 gene sequence is identical to the ScTOK1 gene in 48.6% and CaTok1p with ScTok1p in 31.4%, which can indicate partially different functions of these transporters. The staining of *C. albicans* strains by diS-C_3_(3) is approximately twice as slow as that of *S. cerevisiae* [15]. The reason for this difference in the rate of staining could be a lower Δψ in *C. albicans* cells relative to *S. cerevisiae* cells. Probably for these reasons, the Δψ reduction and membrane depolarization following the blockage of H^+^-ATPase by DM-11 in *C. albicans* are not recovered by Tok1p activity.

Calcium channels in *S. cerevisiae* have been identified as high-affinity and low-affinity calcium uptake systems (HACS and LACS). The voltage-gated Cch1p [46] and the stretch-activated Mid1p [47] form a complex that defines the HACS, whereas Fig1p is a component of LACS [48]. The homologs of these genes in *C. albicans* were found by Brandt et al. [49]. CaCCH1 has a 38.4% identity to its *S. cerevisiae* homolog while the CaMID1 gene sequence had 36.9% identity to ScMID1 [49]. In *S. cerevisiae*, HACS is activated by low Ca^2+^, whereas LACS activity was only revealed under conditions when HACS was inhibited by rich media and its affinity for Ca^2+^ is 16-fold lower [50]. Brandt et al. [49] observed a similar dependence in *C. albicans*. The perturbation of calcium homeostasis by the influx of Ca^2+^ into *C. albicans* cells leads to their death. This finding has allowed amiodarone (AMD) to be used as an antifungal drug. Maresova et al. [41] and Pena et al. [51] suggested that AMD elicits plasma membrane hyperpolarization by inducing K^+^ efflux from the cells followed by depolarization resulting in the Ca^2+^ influx and loss of cell viability.

We used a high concentration of CaCl_2_ (25 mM) to force the cells to take up Ca^2+^ through the LACS system and to induce membrane hyperpolarization. The measurements with diS-C_3_(3) indicated a Δψ increase (λmax = ~578–9 at ~40 min) almost with the same intensity as in the case of cells with a low concentration of CaCl_2_ (λmax = ~577–8 at ~40 min) (Figure 3A). Di-4-ANEPPS fluorescence showed only a slight red shift in cells with 25 mM CaCl_2_ (control EM: 575 nm; CaCl_2_ treated EM: 577 nm) (Figure 3B). Callahora et al. [52] pointed out that agents that did not produce an efflux of K^+^ also did not produce increased Ca^2+^ uptake, and those that produced K^+^ efflux increased Ca^2+^ uptake. Transport across the plasma membrane in *C. albicans* cells appears to be reversible. A slight red shift of the di-4-ANEPPS fluorescence indicating low hyperpolarization in our CaCl_2_ studies may be due to compensation of the negative charge on the outside of membrane by K^+^ efflux from the cells. On the other hand, when we used AMD we observed a fast Δψ build up (λmax = ~578–9 at 8 min) (Figure 3A) and a red shift of di-4-ANEPPS fluorescence (control EM: 575 nm; AMD treated EM: 580 nm) (Figure 3B) indicating membrane hyperpolarization according to studies on *S. cerevisiae* by other researchers [41,51].

### 3.3. Di-4-ANEPPS Is a Suitable Tool for Fast Measuring of the Influence of Detergents on Δψ

In Figure 1, Figure 2 and Figure 3 we show the validation of the usage of di-4-ANEPPS in Δψ measurements in comparison to already known diS-C_3_(3) dye. The di-4-ANEPPS assay is more rapid and reliable due to lack of toxicity towards *C. albicans* cells (Figure 1D) and unlike diS-C_3_(3), the di-4-ANEPPS Δψ measurements are not interfered with by ABC transporters activity (Figure 1E). Here, we wanted to show the vast potential of the di-4-ANEPPS dye for rapid screening of Δψ in *C. albicans* as a result of cell physiology changes. We selected the influence of amphipathic compounds on *C. albicans’* membranes using three detergents: cationic benzalkonium chloride (BAC), anionic sodium dodecyl sulfate (SDS) and non-ionic Triton X-100. Additionally, for more clear presentation of di-4-ANEPPS fluorescence shifts we used an R-B ratio formula, described in Section 2.5.

The mechanism of antifungal action of commonly used detergents is often not well understood. Kodedova et al. [53] showed that detergents at high concentrations cause membrane permeabilization in *S. cerevisiae* and outflow of cations from the inside of the cell. Permeabilized cells cannot maintain Δψ and there is a massive outflow of cations from the inside of the cell. Gaskova et al. [37] noted that this outflow of cations enhances diS-C_3_(3) binding capacity of the cytosolic components and this leads to a fast increase of λmax. 

SDS is an efficient solubilizer of integral membrane proteins [54]. At low concentrations, SDS increased the permeability of the *S. cerevisiae* plasma membrane, as demonstrated by Kodedova et al. [53], using diS-C_3_(3), whereas at a higher concentration (1.44 mg/mL), SDS caused a very rapid red shift of diS-C_3_(3) indicating fully permeabilized membranes. The intensity of antifungal activity of SDS depends on the time of incubation with cells and the concentration used. After a 30 min treatment with SDS, we observed hyperpolarization of the plasma membrane in a range of 80–320 μg/mL SDS (R-B ratio increase of up to ~1.15 at 320 μg/mL) (Figure 4). PI measurements indicate a 25% permeabilization at a concentration of 80 μg/mL SDS and fully permeabilized cells in higher concentrations (Figure 5).

BAC is a quaternary ammonium compound which has been used in clinical applications since 1935 [55]. Kodedova et al. [53] found that 18 μg/mL BAC caused a red shift in *S. cerevisiae* cells stained with diS-C_3_(3), which indicated partial permeabilization of cells while others have been depolarized. At 0.36 μg/mL BAC the cells were depolarized [53]. Our results with di-4-ANEPPS show a similar trend in BAC interaction with *C. albicans* cells. We observed depolarization of cells in the range of 10–40 μg/mL BAC (R-B ratio drop of up to 0.9 at 40 μg/mL) and hyperpolarization when the concentrations of 80–320 μg/mL BAC were used (R-B ratio increase of up to ~1.25 at 320 μg/mL) (Figure 4). As we show in Figure 5, BAC induced the highest permeability of membranes among the used detergents in the concentration of 20 μg/mL (>90% permeabilized cells) and from a concentration of 40 μg/mL we observed a full permeabilization of cells.

The nonionic detergent Triton X-100 was previously used for permeabilization as a tool for the assay of yeast intracellular enzymes in whole cells [56], but the information on the Triton X-100 effect of yeast plasma membrane is scarce. Using di-4-ANEPPS and PI, we observed the weakest effect of Triton X-100 among the detergents used. Triton X-100 induced a blue shift of di-4-ANEPPS (R-B = 0.95 and 0.9 at 160 and 320 μg/mL, respectively), which means depolarization of the *C. albicans* plasma membrane only at the highest concentrations used (Figure 4). We also observed approximately 50% permeabilization of *C. albicans* cells at 320 μg/mL Triton X-100 (Figure 5).

## 4. Conclusions

In this study, we reported the use of di-4-ANEPPS dye on Δψ measurements of *C. albicans*. For the development of the method, we tested different conditions disturbing ion homeostasis, such as cell aging or de- and hyperpolarising agents (KCl and DM-11; CaCl_2_ and AMD) and compared results with the known diS-C_3_(3) assay. We provided new information on the response of *C. albicans* under those conditions and discussed our data based on the findings reported by other research groups using non-pathogenic *S. cerevisiae.* Due to the advantages of di-4-ANEPPS over diS-C_3_(3), we developed an R-B ratio formula for rapid Δψ calculations and proposed it as a method for detection of *C. albicans* physiology disturbances on the example of the influence of commonly used detergents (SDS, BAC and Triton X-100).

## Figures and Tables

**Figure 1 microorganisms-07-00110-f001:**
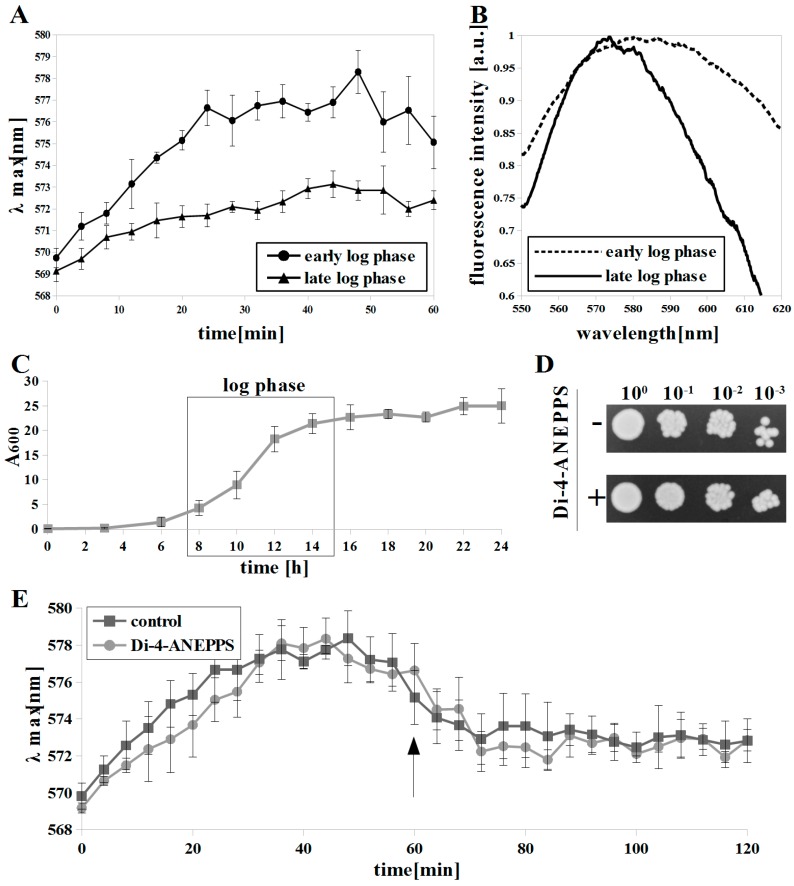
Growth phase-dependent depolarization of *C. albicans* cells measured by: (**A**) monitoring fluorescence maxima in time by diS-C_3_(3), means ± SD (n = 4) and (**B**) fluorescence spectrum shift of di-4-ANEPPS (red-blue signal ratio (R-B) values = 1.043 ± 0.011 and 0.921 ± 0.002 for 8 h and 14 h, respectively), each spectrum is averaged (n = 9); (**C**) growth curve of *C. albicans* CAF2-1, strain was grown to 8 h and 14 h (early and late log phases), means ±SD (n = 3); (**D**) di-4-ANEPPS was not toxic towards *C. albicans* CAF2-1; (**E**) ATP-binding cassettes (ABC) mediated diS-C_3_(3) efflux was not inhibited by addition (**arrow**) of di-4-ANEPPS, means ±SD (n = 4).

**Figure 2 microorganisms-07-00110-f002:**
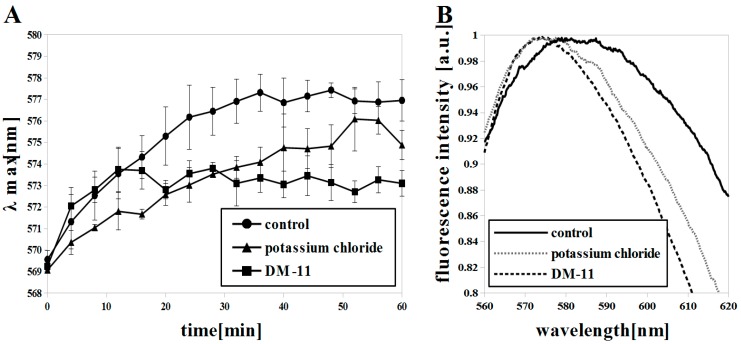
Plasma membrane potential (Δψ) reduction and membrane depolarization in *C. albicans* induced by KCl (200 mM) and DM-11 (50 μM) in early log phase (8 h) shown by: (**A**) diS-C_3_(3), means ± SD (n = 4) and (**B**) di-4-ANEPPS (R-B values = 1.040 ± 0.025, 0.991 ± 0.002 and 0.978 ± 0.010 for control, KCl, DM-11, respectively), each spectrum is averaged (n = 9).

**Figure 3 microorganisms-07-00110-f003:**
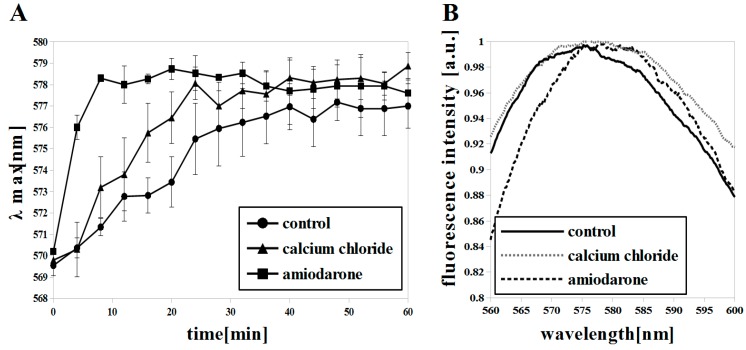
Δψ increase and membrane hyperpolarization in *C. albicans* induced with calcium chloride (CaCl_2_) (25 mM) and amiodarone (AMD) (2 μM) in early log phase (8 h) shown by: (**A**) diS-C_3_(3), means ±SD (n = 4) and (**B**) di-4-ANEPPS (R-B values = 0.978 ± 0.011, 0.996 ± 0.006 and 1.033 ± 0.023 for control, CaCl_2_, AMD, respectively), each spectrum is averaged (n = 9).

**Figure 4 microorganisms-07-00110-f004:**
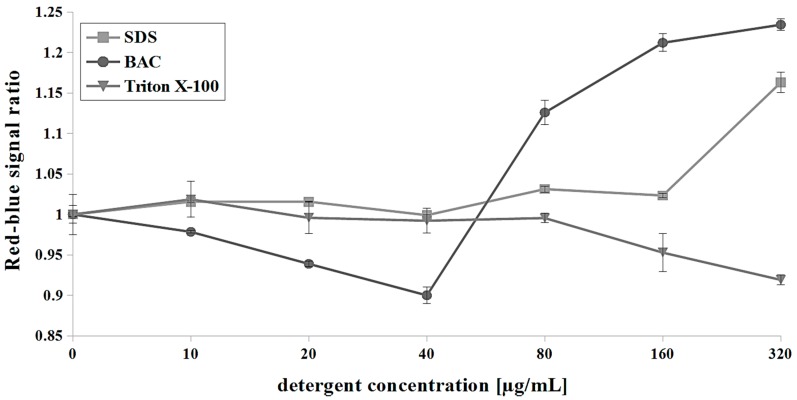
The influence of detergents sodium dodecyl sulfate (SDS) (0–320 µg/mL), benzalkonium chloride (BAC) (0–320 µg/mL) and Triton X-100 (0–320 µg/mL) on Δψ. For data analysis, red-blue signal ratio (R-B ratio) was calculated as described in Section 2.5, means ± SD (n = 9).

**Figure 5 microorganisms-07-00110-f005:**
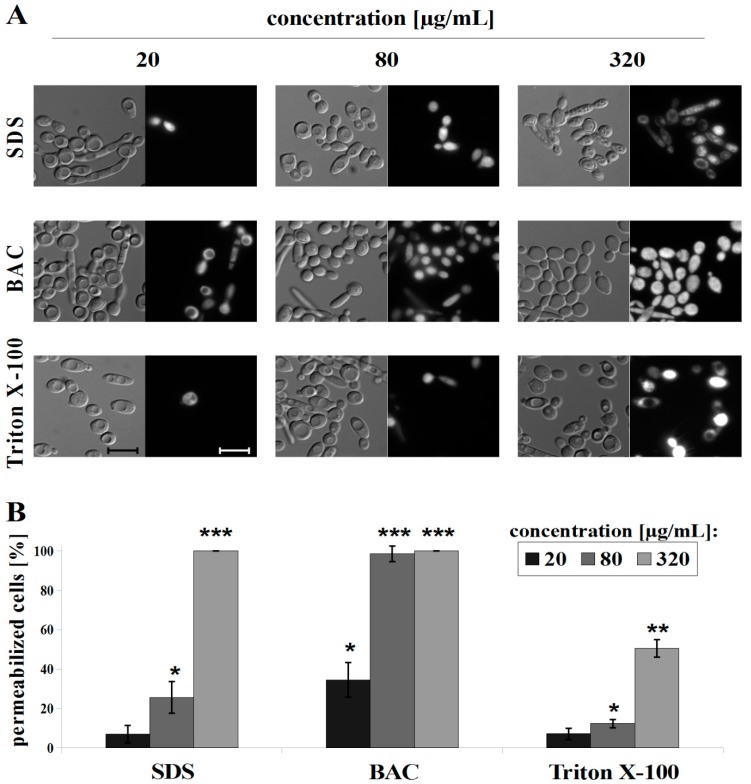
*C. albicans* CAF2-1 strain staining with propidium iodide (PI) after treatment with detergents: SDS, BAC and Triton X-100, presented as: (**A**) microscopic observations, scale bar = 10 µm and (**B**) histograms of the counted % of permeabilized cells, means ± SD (n = 3), statistical analysis at each concentration was performed relative to control experiment without detergent (* *p* < 0.05; ** *p* < 0.01; *** *p* < 0.001).

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
