# Peer review of "Plasma Membrane Potential of *Candida albicans* Measured by Di-4-ANEPPS Fluorescence Depends on Growth Phase and Regulatory Factors"

_microorganisms, 2019, doi:10.3390/microorganisms7040110_

Round 1

Reviewer 1 Report

The work done on paper is nicely planned and the results are well analyzed and concluded. The author’s claims meet the high standard of the journal; therefore I will recommend the acceptance of the manuscript.

Author Response

We would like to thank the reviewer for a very careful reading of

our manuscript and his valuable revision.

Reviewer 2 Report

In the current manuscript by Suchodolski and Krasowska, the authors describe an adaptation of method to measure changes in membrane potential for the fungal pathogen Candida albicans. As such the membrane potential changes are critical in organismal biology and require a closer scientific attention than it has achieved so far. Thus, the authors’ work described here is meritorious as it serves to fill the gap in our current knowledge. Despite the merits of their studies, the manuscript (in my opinion) still can be improved in many aspects and have a significant potential to advance the field if improved on certain aspects. I have highlighted my major suggestions here. Additionally I have also highlighted some minor points which the authors should rectify.

1. Throughout the manuscript the authors highlight the differences and similarities in membrane potential in comparison with S. cerevisiae.  While it is okay to cite the previous literature but the authors heavily rely on the previously reported values. For example, in section 3.2 the authors discuss about the differences in comparison to S. cerevisiae. It maybe true what the authors are surmising, however it is equally likely that the differences in their observations are purely due to technical differences across labs and if they perform the experiments using S. cerevisiae in their lab they may or may not see the differences. For these reasons the authors should use S. cerevisiae alongside, as this not only will serve as a control but considering the authors are establishing a new method, it will also serve as a good tool to highlight the species-specific differences, if any.

2. In their figures, the authors plot their data with error bars, however the figure legends do not describe what these error-bars stand for. Are they SEM or SD or confidence intervals? The authors should specify this. Additionally, the authors should specify what each data point indicates? Are the data points describe a mean of multiple technical replicates or are they mean values across independent experiments?

3. In the figures, where the authors have shown spectra for di-4-ANEPS, it is unclear whether the spectra are weighted average of multiple observations or not. The authors should be specific in their description under the figure legends.

4. In the Figure 4, the authors use red-blue ratio as a quantitative parameter. Why don’t the authors use the similar parameter in describing the first 3 figures as well, as this parameter appears to be very effective in driving home the message. The authors can include a sub-figure for this parameter for first 3 figures.

5. The authors should consider measuring membrane potential of yeast versus hyphal forms of Candida albicans. Moreover, the authors can examine whether fluconazole (or other azoles) can alter membrane potential in C. albicans. Having these experiments can significantly increase the impact of the authors’ findings and can drive excitement in the Candida research community to use their method.

Minor points:

1. No description of figure 1C in text. The authors should have a sentence indicating that the log phase for CAF2 was observed during 8 to 12 hours of growth

2. The figure legend for Figures 1B ,2B and 3B should have a short sentence indicating the time at which the spectrum was recorded.

3. Sentence between lines 32-34 needs a citation

4. On line 31, where authors mention that the fungus is part of human micro flora, the author should be more specific in their description. They should mention that it is a normal constituent of the human micro flora at skin, mouth, gastrointestinal and genital tracts.

5. For line 175 where authors mention that the dye does not have side effect, the authors should modify the sentence to keep it more scientific. The authors may wish to use something such as “the dye’s use did not adversely affect the fungal viability”

6. Line 231 mentions “LACS system”. The authors should clarify this acronym to improve readability.

Author Response

First of all, we would like to thank the Editor and the reviewers for a very careful reading of

our manuscript and their valuable comments that have enabled us to improve its present

revised version and to find some mistakes that have slipped our attention in the previous

version. The answers to reviewer's comments are marked in blue. Changes performed to the revised manuscript are highlighted in yellow.

Response to the reviewer 2

Thank you for comments which are very useful for improving our manuscript. Below we introduce our answers to comments.

1.                  Throughout the manuscript the authors highlight the differences and similarities in membrane potential in comparison with S. cerevisiae.  While it is okay to cite the previous literature but the authors heavily rely on the previously reported values. For example, in section 3.2 the authors discuss about the differences in comparison to S. cerevisiae. It maybe true what the authors are surmising, however it is equally likely that the differences in their observations are purely due to technical differences across labs and if they perform the experiments using S. cerevisiae in their lab they may or may not see the differences. For these reasons the authors should use S. cerevisiae alongside, as this not only will serve as a control but considering the authors are establishing a new method, it will also serve as a good tool to highlight the species-specific differences, if any.

Thank you for the interesting suggestion of performing parallel measurements on S. cerevisae and C. albicans in our laboratory. In fact, our laboratory has been cooperating with laboratories of the Czech Academy of Sciences (prof. Karel Sigler) and the Charles University in Prague (prof. Dana Gaskova) for more than ten years. The exchange of experience and scientific internships took place as part of the grants and international agreements. The results obtained in Czech laboratories were confirmed by the experiments on S. cerevisiae' membrane potential measured with fluorescent dyes and were also carried out in our laboratory. Due to the fact that the Czech laboratories were the first to work on S. cerevisiae and the first to publish the results of their experiments, our laboratory, in accordance with good scientific principles, quoted Czech publications (eg in the following headings: 1, 5, 16, 24, 37, 40). The results quoted by us fully coincide with the results obtained in our laboratory, therefore we decided to include in our publication only the results obtained on C. albicans not described so far.

2.              In their figures, the authors plot their data with error bars, however the figure legends do not describe what these error-bars stand for. Are they SEM or SD or confidence intervals? The authors should specify this. Additionally, the authors should specify what each data point indicates? Are the data points describe a mean of multiple technical replicates or are they mean values across independent experiments?

Thank you for your suggestion. We corrected the data in descriptions below Figures. In M&M section (lines 83-84, 105-106, 138-140) we described how many times we repeated experiments.

3.              In the figures, where the authors have shown spectra for di-4-ANEPS, it is unclear whether the spectra are weighted average of multiple observations or not. The authors should be specific in their description under the figure legends.

Appropriate data we introduced in descriptions below Figures.

4.              In the Figure 4, the authors use red-blue ratio as a quantitative parameter. Why don’t the authors use the similar parameter in describing the first 3 figures as well, as this parameter appears to be very effective in driving home the message. The authors can include a sub-figure for this parameter for first 3 figures.

We are very grateful for this comment. Our aim was to validate the method using known depolarization and hyperpolarization agents in Figs. 1-3. So we think that showing spectral shifts is more appropriate in this approach. In our opinion counting R-B values is suitable in monitoring plasma membrane potential changes in case of experiments with many variable data for better visualisation on the graph as we shown on Fig. 4. For better uniformization of all paper we added R-B values in descriptions to first 3 figures as Reviewer suggested.

5.                  The authors should consider measuring membrane potential of yeast versus hyphal forms of Candida albicans. Moreover, the authors can examine whether fluconazole (or other azoles) can alter membrane potential in C. albicans. Having these experiments can significantly increase the impact of the authors’ findings and can drive excitement in the Candida research community to use their method.

The idea of using the method described by us for measure membrane potential in C. albicans hyphae as well as for examination of fluconazole altering membrane potential is very interesting. However, when we tried to carry out such experiments, we encountered a number of difficulties related, for example, to the heterogeneous dispersion of yeast hyphae in solutions, and thus we obtained unreliable and non-reproducible results. Perhaps our method can be used only in the case of yeast forms of C. albicans.

We are continuing experiments with the effect of fluconazole and other antifungal drugs on the membrane potential of C. albicans using clinical strains. Our intention was to show the basics of the method using the factors directly affecting the cell membrane. Application of the method to a wider understanding of the changes occurring in the membranes of strains isolated from patients will be the subject of our next research.

We want to continue the experiments and hope to find solutions to the problems. We have decided to publish the results obtained so far and we hope that they will open the way to new knowledge about the physiology of C. albicans.

Minor points:

1.              No description of figure 1C in text. The authors should have a sentence indicating that the log phase for CAF2 was observed during 8 to 12 hours of growth

We added the appropriate description in lines 161-162 and 163.

2.              The figure legend for Figures 1B, 2B and 3B should have a short sentence indicating the time at which the spectrum was recorded.

We added the appropriate description in Figures’ captions.

3.              Sentence between lines 32-34 needs a citation

We added a citation in line 35.

4.              On line 31, where authors mention that the fungus is part of human micro flora, the author should be more specific in their description. They should mention that it is a normal constituent of the human micro flora at skin, mouth, gastrointestinal and genital tracts.

We added additional information. Thank you for suggestion.

6.              For line 175 where authors mention that the dye does not have side effect, the authors should modify the sentence to keep it more scientific. The authors may wish to use something such as “the dye’s use did not adversely affect the fungal viability”

We modify our sentence in line 178 according to Reviewers’ suggestion.

7.              Line 231 mentions “LACS system”. The authors should clarify this acronym to improve readability.

We added some sentences about LACS system and we hope that the text is more clear now.

Round 2

Reviewer 2 Report

The authors adequately addressed my concerns.